# Improving Molecular Modeling with Geometric GNNs: an Empirical Study

**Ali Ramlaoui** [* 1 2]   **Théo Saulus** [* 1 2]   **Basile Terver** [* 3 2]
**Victor Schmidt** [4 5 6]   **David Rolnick** [5 7]   **Fragkiskos D. Malliaros** [1 8]   **Alexandre Duval** [1 8 5 6]

## Abstract

Rapid advancements in machine learning (ML) are transforming materials science by significantly speeding up material property calculations. However, the proliferation of ML approaches has made it challenging for scientists to keep up with the most promising techniques. This paper presents an empirical study on Geometric Graph Neural Networks for 3D atomic systems, focusing on the impact on performance, scalability, and symmetry enforcement of different (1) canonicalization methods, (2) graph creation strategies, and (3) auxiliary tasks. Our findings and insights aim to guide researchers in selecting optimal modeling components for molecular modeling tasks. Our code is available at `https://github.com/RolnickLab/ocp`.

## 1. Introduction

The field of computational materials science has witnessed an increasing interest in recent years, with the explosion of machine learning approaches to model material properties at the quantum scale. This is possible thanks to the release of large-scale datasets such as OC20 (Chanussot et al., 2020) and QM7-X (Hoja et al., 2021), which contain millions of molecular structures along with various properties (forces, energy, band gap) computed using quantum mechanical simulations involving Density Functional Theory (DFT) (Kohn et al., 1996).

ML models have been trained to approximate DFT, thus constituting an even faster proxy to the Schrödinger equation, modeling atomic interactions and systems' behavior.

*Equal contribution  [1]Université Paris-Saclay, Centrale-Supélec  [2]École Normale Supérieure Paris-Saclay  [3]École Polytechnique, IP Paris  [4]Université de Montréal  [5]Mila - Quebec AI Institute  [6]Entalpic  [7]McGill University  [8]Inria Saclay. Correspondence to: Ali Ramlaoui <ali.ramlaoui@student-cs.fr>, Théo Saulus <theo.saulus@student-cs.fr>, Basile Terver <basile.terver@polytechnique.edu>.

*Accepted at the 1st Machine Learning for Life and Material Sciences Workshop at ICML 2024*. Copyright 2024 by the author(s).

They enable the calculation of material properties in seconds instead of hours or days, offering the potential to accelerate scientific discovery via high-throughput screening of novel materials (Batatia et al., 2023a; Merchant et al., 2023). However, the explosion of ML approaches in recent years makes it hard to keep up with promising techniques for scientists in the field. While some surveys have attempted to structure the different categories of ML approaches (Han et al., 2022; Duval et al., 2024), these do not focus on empirical evaluation.

In this paper, we propose an empirical study of some key modeling aspects of Geometric GNNs for 3D atomic systems. Specifically, we investigate the impact of 1) the recent canonicalization methods used to enforce or approximate Euclidean symmetries, 2) the graph creation step when modeling a 3D atomic system, and 3) adding several auxiliary tasks. We focus on the OC20 dataset modeling the relaxed adsorption energy of an adsorbate-catalyst system. We hope that the conclusions and insights drawn from our experiments will benefit the community, making it possible to quickly choose the right modeling component.

## 2. Choice of Canonicalization

A function $f : \mathcal{X} \to \mathcal{X}$ is said to be equivariant with respect to a transformation $t$ if $\forall x \in \mathcal{X}, f(t(x)) = t(f(x))$. In particular, $f$ is $E(3)$-equivariant if it is equivariant to rotations, translations, and reflections. $E(3)$-equivariance is a desirable property in molecular modeling to learn representations that are best suited for physically meaningful tasks, such as force predictions on atoms (e.g., S2EF task of OC20). This can be enforced in the architecture of the model during the message passing steps by using equivariant features of the input's representation (Schütt et al., 2021; Batatia et al., 2023b; Liao & Smidt, 2023), which comes at the cost of expensive feature computations. A recent alternative to these equivariant architectures lies in unconstrained GNNs, which do not enforce $E(3)$-equivariance by model design but instead with a coordinate-preprocessing step referred to as canonicalization (Hu et al., 2021; Duval et al., 2023; Pozdnyakov & Ceriotti, 2024). This process grants unconstrained GNNs with greater flexibility, scalability, and often expressivity (Duval et al., 2024). Commonly, it involves

projecting (e.g., with PCA or an equivariant network) the input atomic system into a canonical space such that every Euclidean transformation of the same system gets mapped to the same canonical representation, i.e. handling symmetries in the data pre-processing.

Since these canonicalization methods are all recent and no comparison has yet been drawn, we benchmark the proposed approaches on QM9 and OC20 tasks, evaluating their impact on performance and symmetry enforcement.

## 2.1. Canonicalization for direct predictions

In this subsection, we evaluate several canonicalization procedures with the FAENet backbone architecture (Duval et al., 2023), a powerful unconstrained GNN which is not equivariant by itself. We assume familiarity of the reader with these approaches but provide a description in Appendix A.

- Vector Neurons Network (VNN) (Deng et al., 2021) using the VN re-implementation of PointNet (Qi et al., 2017) and DGCNN (Wang et al., 2019). This class of canonicalization networks is $SO(3)$-equivariant by design and is applied to obtain a canonical representation of the data following the method of Kaba et al. (2023).

- Stochastic Frame Averaging (SFA) (Duval et al., 2023), designed to avoid averaging the predictions over 8 elements of the frame as required by Frame Averaging (Puny et al., 2021). Instead, we sample one canonical orientation at random at each epoch, similar to data augmentation on a small and complete set.

- A novel method denoted SFA+SignNet, where we propose to map the several projection matrices of SFA to a single one using a sign-equivariant network proposed in SignNet (Lim et al., 2023b;a). The rationale is to handle the sign ambiguity problem of PCA that exists in SFA with a small dedicated network to get a unique canonical representation of Euclidean transformations. We propose two implementations of SignNet, either using VNNs to have a perfect $E(3)$-equivariance when combined with SFA or using MLPs without theoretical guarantees.

For VNNs and SFA+SignNet methods, we test both training and freezing the weights of the Canonicalization Network (CN). We report performance on the OC20 IS2RE, OC20 S2EF, and QM9 in Tables 1, 5, 6, 7, 8, and 9.

In our OC20 IS2RE experiment, we found there are almost no differences between the various canonicalization methods, with MAE of 594, 598, and 593 for SFA, VN-PointNet, and VN-DGCNN. Specially, non-exact canonicalizations (SFA and SFA+SignNet) demonstrate equal or

| Canonicalization | Cano. trained parameters | avg. MAE (meV) ↓ | EwT (ID) (%) ↑ | 3D Rotation Invariance ↓ |
|---|---|---|---|---|
| SFA | 0 | 594 | 4.40 | $1.30 \cdot 10^{-2}$ |
| (U) SFA+MLP-SignNet | 0 | **580** | 4.48 | $9.71 \cdot 10^{-2}$ |
| (T) SFA+MLP-SignNet | 454 | 583 | 4.46 | $4.00 \cdot 10^{-2}$ |
| (U) SFA+VN-SignNet | 0 | 592 | **4.69** | $7.58 \cdot 10^{-3}$ |
| (T) SFA+VN-SignNet | 2,620 | 599 | 4.25 | $2.57 \cdot 10^{-2}$ |
| (U) VN-Pointnet | 0 | 605 | 4.09 | $4.62 \cdot 10^{-3}$ |
| (T) VN-Pointnet | 1,310 | 598 | 4.12 | $\mathbf{3.80 \cdot 10^{-3}}$ |
| (U) VN-DGCNN | 0 | 600 | 4.31 | $3.11 \cdot 10^{-2}$ |
| (T) VN-DGCNN | 663,804 | 593 | 4.42 | $9.10 \cdot 10^{-3}$ |

*Table 1.* Invariance comparison of canonicalization methods on OC20 IS2RE dataset. (U) (resp. (T)) indicates an untrained (resp. trained) canonicalization network. FAENet backbone has 4,147,731 parameters. More details in Tables 5 and 6.

better MAE than perfectly equivariant methods (e.g., VN-based). This suggests that heuristics approximation of equivariance should be sufficient in some practical applications like OC20. This is aligned with what Duval et al. (2023) suggested when showing that SFA outperforms exact Frame Averaging in terms of downstream performance.

In terms of symmetry enforcement, non-exact methods are surprisingly nearly as effective as fully invariant methods, suggesting that the FAENet backbone implicitly learns to handle symmetries.

Regarding exact canonicalization methods, we observe that training or not the network and swapping one method for another has little impact on model performance. This tends to indicate that the canonical networks's ability to introduce equivariance is more critical than the choice of the canonicalization method.

## 2.2. Canonicalization for relaxed IS2RE

Previous work showed that solving the IS2RE task yields better results by performing relaxed energy predictions rather than direct energy estimation (Liao et al., 2023). Here, we evaluate whether canonicalization methods also perform well at relaxing a trajectory. Table 2 reports the performances of FAENet with multiple symmetry-preserving methods, with the invariant SchNet model and direct IS2RE acting as baseline. Our findings suggest that relaxed IS2RE predictions are competitive with direct IS2RE predictions and are interesting directions to explore for improving molecular property predictions with these architectures. Moreover, a potential explanation as to why the relaxations do not yield significant improvements over direct IS2RE may involve the approximate equivariance or the lack of continuity in some of these canonicalization methods, as pointed out in Dym et al. (2024), which may hamper accurate and smooth relaxations trajectories. This is mainly true for SFA, where a frame is randomly picked at each step of the relaxation, meaning that the canonical inputs can also be far from each other. Lastly, note that exact equivariant meth-

| Model | IS2RE | | IS2RS | |
|---|---|---|---|---|
| | EwT (%) ↑ | MAE (eV) ↓ | DwT (%) ↑ | Pos. MAE ↓ |
| FAENet (Direct) | 4.05 | 0.551 | - | - |
| FAENet (SFA) | 4.92 | 0.587 | 31.1 | 0.390 |
| FAENet (UTPN) | 5.64 | 0.560 | 33.7 | 0.381 |
| SchNet | 1.89 | 0.912 | 15.0 | 0.461 |

*Table 2.* Results on the IS2RE and IS2RS tasks for the VAL-ID validation dataset of OC20 using iterative relaxations. The S2EF models' results are reported in Appendix A.5. Note that these results can only be obtained by keeping the tag 0 atoms of the subsurface as they are important for relaxations.

| Model | ID | |
|---|---|---|
| | EwT (%) ↑ | MAE (eV) ↓ |
| Cutoff 30 - Max neighbours 40 | 2.65 | 0.697 |
| Cutoff 20 - Max. neighbours 40 | 3.08 | 0.673 |
| Cutoff 20 - Max. neighbours 10 | 2.25 | 0.768 |
| Cutoff 10 - Max. neighbours 50 | 4.17 | 0.553 |
| Cutoff 10 - Max. neighbours 10 | 4.49 | 0.553 |
| Cutoff 6 - Max. neighbours 40 | 4.31 | 0.553 |
| Cutoff 1 - Max. neighbours 40 | 1.35 | 1.069 |

*Table 3.* Impact of the cutoff on the performance of FAENet on the OC20 IS2RE task. Full Table in Appendix B.1.

ods for relaxed IS2RE predictions such as the untrained VN-PointNet (UTPN) implementation from (Kaba et al., 2023) yields better MAE than the approximate equivariance module SFA. Still, during our experiments we found that with further focus on the accuracy of S2EF models, canonicalization methods used to enforce symmetries can become more appealing using iterative relaxation methods.

## 3. Graph Creation Study

For large molecular structures, electrostatic long-range interactions are non-negligible components of the system's dynamics (Gasteiger et al., 2019). While various methods have tried to model long-range interactions between faraway atoms, they often suffer from the over-smoothing effect when increasing the number of interaction layers (Liao & Smidt, 2023). As a result, models where the geometric graph is modelled using a cutoff distance between neighbour atoms have been shown to work best. Since properly handling these interactions is essential to accurately simulate the system, we explore in this section the impact of this creation step and the rewiring strategies.

### 3.1. Graph cutoff and rewiring

First, we vary the cutoff distance[1] used during the creation of the graph to check whether linking more atoms with each other helps in learning these couplings. We report the results for the FAENet model on the OC20 IS2RE task in Table 3.

A small cutoff of 1.0 Å leads to the weakest performance, which makes sense since the associated graph is almost empty and nodes are isolated, i.e. messages can not pass correctly. A large cutoff or a complete graph also leads to poor performance despite every atomic interaction being modeled. Within an intermediary range of cutoff values, the model achieves optimal learning with computational efficiency. This is in accordance with past observations where the locality bias of GNN models seemed to fit really well with atomic system property prediction tasks. Thus,

---

[1]When representing the 3D point cloud with a graph, we create an edge between any two atoms if their are within a fixed cutoff distance, and no edge otherwise.

although the creation of the graph through a well-chosen cutoff is important, the margin for fine-tuning this parameter is large enough.

The fact that better graphs are adapted to GNN functioning rather than the precise modeling of the situation, where all atoms interact with each other, echoes what Duval et al. (2022) have stated. Indeed, they showed that to fit the GNN message passing scheme, removing repeating subsurface atoms of the adsorbate did not affect the model performance on IS2RE tasks. Similarly to having a moderate cutoff, the performance improvements of such a strategy are decisive for scalability. As an empirical example, we tried to run an EquiformerV2 (Liao et al., 2023) model for IS2RE with the remove-tag-0 method on an 80GB A100 GPU, leading to a 5× speed-up with no performance loss and proving the relevance of this technique even on very large models.

### 3.2. Ewald-based long range message passing

Since a small value for the cutoff seems to be the most interesting one, we want to model the long interactions differently than adding links between all atoms. Ewald-based Message Passing (EMP) (Kosmala et al., 2023) is introduced in this perspective. It incorporates a physics-based prior in the architecture to model the long-range electrostatic potential via a nonlocal Fourier space scheme, drawing edges based on a cutoff on frequency instead of distances.

Our experiments, given in Table 4, show that EMP is interesting for an invariant method like SchNet (Schütt et al., 2017), which limits its geometric information to atom pairwise distances. However, EMP does not benefit more advanced GNNs like FAENet. To understand why, we plot in Figure 1 the cosine similarity between the embeddings throughout interaction layers. We observe that SchNet and FAENet learn very different representations. Indeed, while the representation of each atom in SchNet tends to be similar to nearby atoms, FAENet is able to give very different embeddings for them. A potential explanation could be that because FAENet is a more expressive model, the propagated messages are less constrained and thus can lead to very diverse atom representations without Ewald. On the other hand, embeddings of SchNet only become diversified

| Model | ID | |
|---|---|---|
| | EwT (%) ↑ | MAE (eV) ↓ |
| FAENet (Graph Rewiring) | 4.05 | 0.551 |
| FAENet (Graph Rewiring) + Ewald | 4.12 | 0.562 |
| FAENet (No Graph Rewiring) | 4.54 | 0.544 |
| FAENet (No Graph Rewiring) + Ewald | 4.11 | 0.556 |
| SchNet (Graph Rewiring) | 3.18 | 0.641 |
| SchNet (Graph Rewiring) + Ewald | 3.54 | 0.604 |
| SchNet (No Graph Rewiring) | 2.93 | 0.654 |
| SchNet (No Graph Rewiring) + Ewald | 3.48 | 0.597 |

*Table 4.* Comparison of the performances of FAENet and SchNet with and without Ewald Message Passing on the IS2RE task. Full table and results for the QM9 dataset in Appendix B.2.

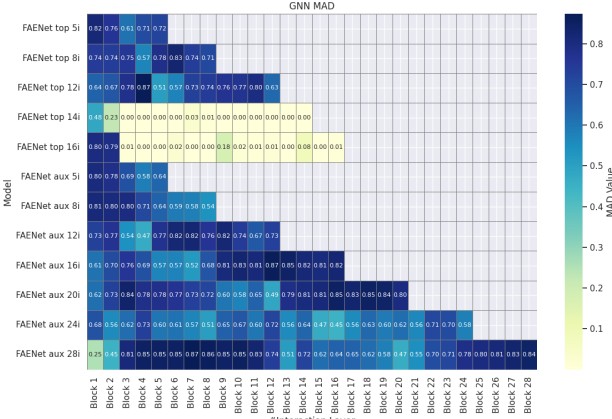

*Figure 2.* MAD values of the graph embeddings (averaged over 50 randomly sampled graphs of the train set) throughout the interaction layers for various models. "FAENet top" models are trained with the top configs of the classical FAENet model (Duval et al., 2023) but with more epochs and lower batch size (128). "FAENet aux" models are our models trained on IS2RE with IS2RS auxiliary task. A model having xx interaction layers is indicated as "XXi".

using Ewald, maybe because incorporating longer messages helps create more distinct representations. More plots can be found in Appendix B.2. One interesting takeaway is that EMP helps take into account long-range interactions for simple models and GNNs with symmetry-constrained layers (Kosmala et al., 2023) but is less efficient on expressive models.

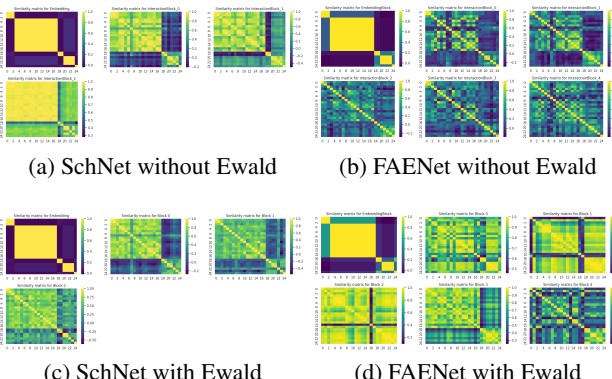

(a) SchNet without Ewald    (b) FAENet without Ewald

(c) SchNet with Ewald    (d) FAENet with Ewald

*Figure 1.* Similarity matrix of the embeddings of the atoms of a randomly picked system for different interaction blocks from the training set of OC20. The same system is used every time to be able to compare the different results.

## 4. Auxiliary Tasks

In this section, we study how to leverage other tasks to improve the performance of FAENet on IS2RE.

### 4.1. Noisy nodes IS2RS auxiliary task

First, we recall that increasing the number of interaction blocks above 8 does not yield better performance for most geometric GNNs, with a dramatic loss of information after 14 layers in the classical FAENet IS2RE setup, as illustrated in Table 14. To address the oversmoothing issue, Godwin et al. (2022) propose to use noisy regularisation, introducing their method called Noisy Nodes. It consists of adding an auxiliary node-level denoising task that encourages diversity

in the latent representations of the nodes (more details in C.2).

Here, we implement Noisy Nodes for the IS2RE task, which is done by adding a position decoding head running in parallel to the original energy prediction head. Our implementation takes inspiration from other state-of-the-art models such as EquiformerV1 (Liao & Smidt, 2023), which benefit from using Noisy Nodes on the IS2RE task (more details in C.2).

To understand the reason for the performance drop for the classical FAENet IS2RE setup of (Duval et al., 2023) when adding interaction layers in Table 14, we plot the Mean Average Distance (MAD) (Chen et al., 2019), averaged over 50 input adsorbate-catalyst pairs, of their embeddings throughout the interaction blocks in Figure 2. The classical FAENet models ("FAENet top") with 14 interaction layers or more see their latent node representations all collapse to almost the same vector, since the MAD goes to almost zero as we go deeper in the model's interaction layers, which is a manifestation of oversmoothing. Figure 2 shows that the models trained with Noisy Nodes IS2RS auxiliary task ("FAENet aux") do not suffer from oversmoothing (i.e. MAD going to zero) even when going as deep as 28 interaction layers.

Then, we compare the performances of our new FAENet models trained on IS2RE Noisy Nodes, varying the number of interaction layers, as summarized in Table 15. First, we observe a clear correlation between the performances and the number of blocks. Second, the best model trained with Noisy Nodes, which has 26 interaction blocks, outperforms

the best models of (Duval et al., 2023), allowing the average MAE to decrease from 568 meV to 525 and the average EwT percentage to increase from 3.78% to 4.43%. Details about our experimental setup are given in Appendix C.1. We observe that the throughput at inference is divided by two between the smallest (5 interaction layers) and biggest model (28 layers). The best tradeoff seems to be around 16 interaction layers, similar to what (Liao & Smidt, 2023) choose for their models trained with IS2RS auxiliary task.

The current most general state-of-the-art models for atomic property prediction, such as (Shoghi et al., 2024) only make use of backbone GNNs (such as GemNet-OC (Gasteiger et al., 2022)) with not more than 6 interaction blocks. Noisy Nodes proves to be a regularisation technique that can be used across models, having equivariant (Liao & Smidt, 2023) or non-equivariant (Duval et al., 2023) features, to leverage the power of more interaction layers and improve performance. On direct IS2RE, our experiments only show a slightly increasing performance with the number of blocks. Yet, unlocking the potential of deeper GNNs with this simple regularisation allows greater freedom in training or pretraining on larger datasets or for more complex tasks, such as relaxed IS2RE 2.2. Our focus for FAENet was on IS2RE, but we expect similar results on S2EF, IS2RS, and other datasets such as QM9 or QM7-X, as obtained by (Godwin et al., 2022) with other GNNs such as GNS (Sanchez-Gonzalez et al., 2020).

### 4.2. Learning equivariance with more interaction layers

As previously stated, equivariant networks are preferred when handling 3D point clouds. However, in the field of protein structure prediction, we were caught by the AlphaFold3 architecture (Abramson et al., 2024) that chose non-equivariant networks in its diffusion module, contrary to AlphaFold2 (Jumper et al., 2021). Thus, we tested whether a deep GNN with no equivariance enforced could learn the equivariance from the data and match performances with canonicalized models using the FAENet backbone, as in Section 2.

We compare our models trained on IS2RE with IS2RS auxiliary task in two settings: with SE(3)-SFA and without any canonicalization method (No-FA). The results are displayed in Table 16. They reveal that, even with many additional interaction blocks (up to 26), both the MAE and the equivariance property do not improve compared to using SE(3)-SFA, meaning that imposing equivariance is still a beneficial inductive bias. Unfortunately, in our setting, having a deeper network does not allow to learn invariance and equivariance more effectively. Further analysis would be needed with larger datasets and other methods to reinforce this argument or find its limits (if any).

### 4.3. Pre-training on different tasks

With the release of larger datasets, the community has increasingly shifted towards pre-training and transfer learning approaches (Batatia et al., 2023a; Shoghi et al., 2024; Deng et al., 2023). We make a step in this direction in this subsection by pre-training our model on the S2EF, which contains roughly two orders of magnitude more data points, and by fine-tuning it on IS2RE, hoping to transfer some knowledge of atomic interactions. More precisely, we leverage the extensive S2EF dataset to utilize all trajectories while maintaining high inference throughput by performing direct energy predictions. Although training on S2EF is more time-consuming than on IS2RE, separating the learning of relaxation and molecular interactions may yield better results than training IS2RE directly from scratch.

Figure 7 in Appendix C.6 shows that with this approach, the energy MAE starts at a better value during training but evolves much slower and converges to a slightly better result (0.53 eV vs. 0.55 eV on the validation ID split). This shows that while the model starts with a good representation of molecular dynamics, the differences are not significant because dynamics are not taken into account by the S2EF training process. Learning molecular interactions through auxiliary tasks with or without dynamics is helpful to achieve better performances, but this pre- or joint training needs to be correctly incorporated into the architecture so as not to overwrite the learned information during the downstream task. This opens the way for new architectures designed to leverage materials design knowledge between tasks and datasets (Shoghi et al., 2024). Whether by training with auxiliary tasks or on other datasets, there seems to be transfer learning and generalization capabilities in atomic property prediction, as in NLP. Hence, we recommend to further explore this promising area of research.

## 5. Conclusion

In this study, we explored various techniques aimed at enhancing the performance of geometric GNNs for molecular modeling. Our empirical study covered several aspects, with the main observations being summarized below.

**Canonicalization methods.** While exact methods provide the best theoretical guarantees for equivariance, approximative heuristics such as SFA seem to yield better performance. This opens questions about how to design canonicalizations that are the most effective in practice, beyond theoretical guarantees, and leaves a broader set of possibilities for the model design, too.

**Graph creation.** Although accurate graph construction is important, many viable options can be considered without significant differences in performance. Furthermore,

physics-inspired modules such as Ewald-based message passing demonstrate improved performance for symmetry-constrained models such as SchNet but do not provide any benefits to more expressive models such as FAENet.

**Auxiliary tasks.** Implementing Noisy Nodes as an auxiliary task significantly enhances the performance of FAENet by leveraging the benefits of much deeper GNNs. As with pretraining on different tasks such as S2EF, there is evidence of transfer learning for atomic property prediction, and we recommend more exploration of this path in the flavor of (Shoghi et al., 2024).

Future work could focus on refining these techniques, exploring their applications across a wider spectrum of datasets, and developing new methods to combine the strengths of various approaches.

## Acknowledgement

The authors thank Sékou-Oumar Kaba, Derek Lim, Joshua David Robinson, and Hannah Lawrence for their insightful comments and discussions, as well as the anonymous reviewers for their suggestions and feedback. Supported in part by ANR (French National Research Agency) under the JCJC project GraphIA (ANR-20-CE23-0009-01) and the Canada CIFAR AI Chairs program. This research was enabled in part by computing resources provided by Mila (mila.quebec) and material support from NVIDIA Corporation in the form of computational resources.

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
