# A. Canonicalization

## A.1. Stochastic Frame Averaging

**Frame Averaging**   We recall the idea of Frame Averaging introduced by Puny et al. (2021), in which $\mathcal{X}$ and $\mathcal{Y}$ denote normed linear space with respective representations $\rho_1$ and $\rho_2$ of a group $G$. In our case, the group of interest is $E(3)$. A *frame* is defined as a function $\mathcal{F} : \mathcal{X} \to 2^G$ taking values in a non-empty subset of the group $G$ such that it is $G$-equivariant and bounded. Under such conditions, and for every $\Phi : \mathcal{X} \to \mathcal{Y}$, the function $\langle\Phi\rangle_{\mathcal{F}} : \mathcal{X} \to \mathcal{Y}$, called *average over the frame* $\mathcal{F}$, and defined as

$$\langle\Phi\rangle_{\mathcal{F}} : x \mapsto \frac{1}{|\mathcal{F}(x)|} \sum_{g \in \mathcal{F}(x)} \rho_2(g)\Phi(\rho_1(g)^{-1}x), \tag{1}$$

is $G$-equivariant. This allows to create an arbitrary neural network, and guarantee the symmetry by averaging the outputs over a well-chosen frame.

**Choosing the frame.**   As in Duval et al. (2023), the neural network takes as input $X$ the atoms position, $Z$ is their atomic numbers, and has outputs in $\mathcal{Y}$, where $\mathcal{Y} = \mathbb{R}^{n \times 3}$ in the case of force predictions, and $\mathcal{Y} = \mathbb{R}$ in the case of energy prediction. The group $E(3)$ only acts on $X$, and not $Z$. A Principal Component Analysis (PCA) on the atomic structure allows to decompose the covariance matrix of the points cloud $\Sigma = U^T \Lambda U$ derived from the centroid of the positions $t = \frac{1}{n}X^T\mathbf{1}$, with $\Lambda$ the diagonal matrix containing the three eigenvalues $\lambda_1 > \lambda_2 > \lambda_3$ (assumed distinct because we consider non-planar structures). The frame is taken as

$$\mathcal{F}(X) = \{(U, t) \mid U = [\pm u_1, \pm u_2, \pm u_3]\}, \tag{2}$$

which is a subset of $E(3)$. The authors prove that the frame $\mathcal{F}$ defined as such is $G$-equivariant and bounded.

**Stochastic Frame Averaging.**   This process requires to average the predictions of the neural network over $|\mathcal{F}(X)| = 2^3 = 8$ elements of the frame. In order to make the computations faster, Duval et al. (2023) sample only one element from the frame instead of performing the average. Although Stochastic Frame-Averaging (SFA) does not have theoretical guarantees, it has been experimentally shown to learn almost perfect equivariance.

## A.2. SFA+SignNet

Initially proposed to help spectral graph representation learning, SignNet (Lim et al., 2023b) is a network which outputs are invariant to sign flips. The authors state that a continuous function $\eta : \mathbb{R}^n \to \mathbb{R}^d$ is *sign-invariant* if and only if $\eta(x) = \kappa(x) + \kappa(-x)$ for some (freely chosen) continuous function $\kappa : \mathbb{R}^n \to \mathbb{R}^d$. SignNet $: \mathbb{R}^{n \times k} \to \mathbb{R}^{n \times k}$ is then defined as:

$$\text{SignNet}(x_1, \ldots, x_k) = \mu\left([\kappa(x_i) + \kappa(-x_i)]_{i=1}^k\right), \tag{3}$$

where $\mu$ and $\kappa$ are neural networks chosen freely.

We propose to apply such a network on the sampled element of the frame $U$ and parametrize $\mu$ and $\kappa$ either with MLPs or with VN-PointNets. In order to constrain the output, we further orthonormalize the output with a Gram-Schmidt process:

$$U' = \text{Gram-Schmidt}(\text{SignNet}(U)). \tag{4}$$

When parametrizing SignNet with MLPs, since we apply non-linearities on the orthogonal matrix $U$, there is no theoretical guarantee for the whole process to be $E(3)$-equivariant a priori. This is coherent with empirical observations when using this method, although training $\mu$ and $\kappa$ helps better enforce the equivariance.

When parametrizing SignNet with VN-PointNets, the whole network is made exactly $E(3)$-equivariant. As explained by Lim et al. (2023a) (section 2.2), the matrix $U$ is orthogonal and unique up to sign changes, while the SignNet function is both sign invariant by design and $O(3)$-equivariant thanks to the use of VN-PointNets.

## A.3. Vector Neurons Network

VNNs are a class of $SO(3)$-equivariant models, where usual neurons are replaced with so-called Vector Neurons: for a given layer, non-linearities output a matrix of size $h \times 3$ instead of a vector of length $h$.

Using this framework, Deng et al. (2021) re-implement classic operations such as linear layers, non-linearities, pooling operations, and normalization layers. They also prove that those layers are all $SO(3)$-equivariants, which allows to re-implement classical networks into their VN variant. In particular, they re-implement the VN variants of two well-known networks from the point clouds literature: PointNet (Qi et al., 2017) and DGCNN (Wang et al., 2019), and test them on classification, segmentation, and reconstruction tasks. They show that accuracy increases compared to the classic implementations and that their equivariance property is indeed (almost) perfectly enforced.

To obtain an $O(3)$ (and then $E(3)$) equivariance, the output of the VNN has to be further orthonormalized with a Gram-Schmidt process to canonicalize the representation in $O(3)$, as described by Kaba et al. (2023).

The following Vector Neurons Networks (VNNs) are used:

- VN-Pointnet with a varying number of VNLinearLeakyReLU layers (between 1 and 3), with the implementation of Deng et al. (2021).

- VN-DGCNN, with the implementation is the one of Deng et al. (2021).

To summarize, we use VNNs to learn the transformation $U$ from the positions $X$:

$$\text{VNN} : \mathbb{R}^{n \times 3} \to \mathbb{R}^{3 \times 3}$$
$$X \mapsto U.$$

$U$ is then orthonormalized with a Gram-Schmidt process.

### A.4. Experimental comparisons

OC20 IS2RE: Tables 5 and 6

OC20 S2EF: Tables 7 and 8

QM9: Table 9

In all OC20 experiments of this section, for a fair assessment, SFA is used in 3D mode, i.e. without the computational trick to force the $z$ axis to remain fixed during canonicalization, which is specific to OC20. The same goes for the methods derived from SFA. As a consequence, the reported performances are lower than reported in other sections.

| Canonicalization | Cano. trainable parameters | 2D Rotation ↓ Invariance | 3D Rotation ↓ Invariance | Reflection ↓ Invariance |
|---|---|---|---|---|
| SFA | 0 | $1.29 \cdot 10^{-2}$ | $1.32 \cdot 10^{-2}$ | $1.30 \cdot 10^{-2}$ |
| Untrained SFA+MLP-SignNet | 0 | $1.01 \cdot 10^{-1}$ | $1.00 \cdot 10^{-1}$ | $9.71 \cdot 10^{-2}$ |
| Trained SFA+MLP-SignNet | 454 | $4.21 \cdot 10^{-2}$ | $7.74 \cdot 10^{-2}$ | $4.00 \cdot 10^{-2}$ |
| Untrained SFA+VN-SignNet | 0 | $6.89 \cdot 10^{-3}$ | $7.58 \cdot 10^{-3}$ | $7.37 \cdot 10^{-3}$ |
| Trained SFA+VN-SignNet | 2,620 | $2.45 \cdot 10^{-2}$ | $2.66 \cdot 10^{-2}$ | $2.43 \cdot 10^{-2}$ |
| Untrained VN-Pointnet (2 hid.) | 0 | $4.61 \cdot 10^{-3}$ | $4.62 \cdot 10^{-3}$ | $4.62 \cdot 10^{-3}$ |
| Trained VN-Pointnet (2 hid.) | 1,310 | $\mathbf{3.63 \cdot 10^{-3}}$ | $\mathbf{3.72 \cdot 10^{-3}}$ | $\mathbf{3.80 \cdot 10^{-3}}$ |
| Untrained VN-Pointnet (1 hid.) | 0 | $4.28 \cdot 10^{-3}$ | $4.28 \cdot 10^{-3}$ | $4.29 \cdot 10^{-3}$ |
| Untrained VN-Pointnet (0 hid.) | 0 | $2.76 \cdot 10^{-2}$ | $2.76 \cdot 10^{-2}$ | $2.79 \cdot 10^{-2}$ |
| Trained VN-Pointnet (0 hid.) | 24 | $1.86 \cdot 10^{-2}$ | $2.31 \cdot 10^{-2}$ | $2.36 \cdot 10^{-2}$ |
| Untrained VN-DGCNN | 0 | $3.03 \cdot 10^{-2}$ | $3.08 \cdot 10^{-2}$ | $3.11 \cdot 10^{-2}$ |
| Trained VN-DGCNN | 663,804 | $9.89 \cdot 10^{-3}$ | $2.49 \cdot 10^{-2}$ | $9.10 \cdot 10^{-3}$ |

*Table 5.* Invariance comparison of canonicalization methods on OC20 IS2RE dataset. The FAENet backbone for this task and dataset has 4,147,731 parameters (5 interaction blocks). We measure the rotation invariance and reflection invariance property as the difference in prediction between every samples D1 (of the ID val split) and D2 defined as a SO(3) transformation of D1, in eV.

# Improving Molecular Modeling with Geometric GNNs: an Empirical Study

| Canonicalization | ID | | OOD-CAT | | OOD-ADS | | OOD-BOTH | |
|---|---|---|---|---|---|---|---|---|
| | EwT (%) ↑ | MAE (meV) ↓ | EwT (%) ↑ | MAE (meV) ↓ | EwT (%) ↑ | MAE (meV) ↓ | EwT (%) ↑ | MAE (meV) ↓ |
| SFA | 4.40 | 566 | 4.12 | 563 | 2.56 | 652 | 2.77 | 594 |
| Untrained SFA+MLP-SignNet | 4.48 | **554** | 4.46 | 552 | 2.75 | **637** | 2.88 | **576** |
| Trained SFA+MLP-SignNet | 4.46 | **554** | 4.51 | 551 | 2.67 | 642 | 2.78 | 586 |
| Untrained SFA+VN-SignNet | **4.69** | 563 | 4.60 | 558 | 2.62 | 651 | 2.59 | 595 |
| Trained SFA+VN-SignNet | 4.25 | 572 | 4.27 | 568 | **2.92** | 658 | **2.97** | 596 |
| Untrained VN-Pointnet (2 hid.) | 4.09 | 567 | **4.66** | 565 | 2.60 | 673 | 2.85 | 615 |
| Trained VN-Pointnet (2 hid.) | 4.12 | 568 | 4.33 | 563 | 2.77 | 658 | 2.75 | 604 |
| Untrained VN-Pointnet (1 hid.) | 4.37 | 565 | 4.20 | 561 | 2.64 | 666 | 2.73 | 614 |
| Untrained VN-Pointnet (0 hid.) | 4.01 | 581 | 3.92 | 571 | 2.75 | 660 | 2.64 | 615 |
| Trained VN-Pointnet (0 hid.) | 4.14 | 567 | 4.36 | 563 | 2.56 | 675 | 2.88 | 614 |
| Untrained VN-DGCNN | 4.31 | 567 | 4.14 | 562 | 2.58 | 660 | 2.72 | 610 |
| Trained VN-DGCNN | 4.42 | 560 | 4.40 | 556 | 2.78 | 656 | 2.81 | 601 |

*Table 6.* Performance comparison of canonicalization methods on OC20 IS2RE dataset. All models were trained for 12 epochs using Duval et al. (2023) config.

| Canonicalization | Cano. trainable parameters | Energy invariance | | Forces equivariance | |
|---|---|---|---|---|---|
| | | 3D Rotation ↓ | Reflection ↓ | 3D Rotation ↓ | Reflection ↓ |
| SFA | 0 | $1.88 \cdot 10^{-2}$ | $1.88 \cdot 10^{-2}$ | $7.17 \cdot 10^{-2}$ | $8.34 \cdot 10^{-3}$ |
| Untrained SFA+MLP-SignNet | 0 | $7.81 \cdot 10^{-2}$ | $7.61 \cdot 10^{-2}$ | $7.44 \cdot 10^{-2}$ | $2.04 \cdot 10^{-2}$ |
| Trained SFA+MLP-SignNet | 454 | $6.57 \cdot 10^{-2}$ | $3.57 \cdot 10^{-2}$ | $7.35 \cdot 10^{-2}$ | $1.17 \cdot 10^{-2}$ |
| Untrained SFA+VN-SignNet | 0 | $2.07 \cdot 10^{-2}$ | $2.04 \cdot 10^{-2}$ | $6.86 \cdot 10^{-2}$ | $9.42 \cdot 10^{-3}$ |
| Trained SFA+VN-SignNet | 2,620 | $1.92 \cdot 10^{-2}$ | $1.89 \cdot 10^{-2}$ | $6.55 \cdot 10^{-2}$ | $8.72 \cdot 10^{-3}$ |
| Untrained VN-Pointnet (2 hid.) | 0 | $1.80 \cdot 10^{-2}$ | $1.80 \cdot 10^{-2}$ | $6.92 \cdot 10^{-2}$ | $8.78 \cdot 10^{-3}$ |
| Trained VN-Pointnet (2 hid.) | 1,310 | $1.67 \cdot 10^{-2}$ | $1.67 \cdot 10^{-2}$ | $6.89 \cdot 10^{-2}$ | $8.77 \cdot 10^{-3}$ |
| Untrained VN-Pointnet (0 hid.) | 0 | $3.50 \cdot 10^{-2}$ | $3.49 \cdot 10^{-2}$ | $6.96 \cdot 10^{-2}$ | $1.08 \cdot 10^{-2}$ |
| Trained VN-Pointnet (0 hid.) | 24 | $3.31 \cdot 10^{-2}$ | $3.34 \cdot 10^{-2}$ | $7.00 \cdot 10^{-2}$ | $1.05 \cdot 10^{-2}$ |
| Untrained VN-DGCNN | 0 | $\mathbf{1.50 \cdot 10^{-2}}$ | $\mathbf{1.50 \cdot 10^{-2}}$ | $\mathbf{6.83 \cdot 10^{-2}}$ | $\mathbf{3.58 \cdot 10^{-3}}$ |
| Trained VN-DGCNN | 663,804 | $2.02 \cdot 10^{-2}$ | $1.50 \cdot 10^{-2}$ | $6.91 \cdot 10^{-2}$ | $8.09 \cdot 10^{-3}$ |

*Table 7.* Equivariance comparison of canonicalization methods on OC20 S2EF dataset. The FAENet backbone for this task and dataset has 5,675,410 parameters (7 interaction blocks). We measure the energy rotation invariance, energy reflection invariance, force rotation equivariance, and force reflection equivariance properties as the difference in prediction between every sample D1 (of the ID val split) and D2 defined as a SO(3) transformation of D1, in eV.

| Canonicalization | Energy MAE (mEV) ↓ | | | | Force MAE (mEV) ↓ | | | |
|---|---|---|---|---|---|---|---|---|
| | ID | OOD Cat | OOD Ads | OOD Both | ID | OOD Cat | OOD Ads | OOD Both |
| SFA | 424 | 445 | 579 | 680 | 55.6 | 55.2 | 63.2 | 74.6 |
| Untrained SFA+MLP-SignNet | **420** | **444** | **515** | **631** | **54.0** | **53.8** | **61.4** | **72.4** |
| Trained SFA+MLP-SignNet | 422 | 446 | 558 | 666 | 54.2 | 53.9 | 62.7 | 73.8 |
| Untrained SFA+VN-SignNet | 439 | 458 | 565 | 673 | 56.5 | 56.0 | 65.3 | 76.7 |
| Trained SFA+VN-SignNet | 442 | 464 | 590 | 701 | 58.0 | 57.5 | 65.2 | 76.9 |
| Untrained VN-Pointnet (2 hid.) | 435 | 455 | 596 | 697 | 56.0 | 55.6 | 66.9 | 77.5 |
| Trained VN-Pointnet (2 hid.) | 435 | 453 | 585 | 696 | 56.1 | 55.8 | 64.2 | 75.6 |
| Untrained VN-Pointnet (0 hid.) | 440 | 459 | 597 | 705 | 56.0 | 55.6 | 64.7 | 75.8 |
| Trained VN-Pointnet (0 hid.) | 440 | 459 | 572 | 671 | 55.8 | 55.4 | 63.7 | 74.9 |
| Untrained VN-DGCNN | 456 | 474 | 593 | 763 | 55.7 | 55.5 | 65.8 | 76.9 |
| Trained VN-DGCNN | 432 | 453 | 662 | 762 | 55.5 | 55.2 | 71.2 | 80.7 |

*Table 8.* Performance comparison of canonicalization methods on OC20 S2EF dataset. All models were trained for 12 epochs using Duval et al. (2023) config.

| Canonicalization | Cano. trainable parameters | MAE (meV) ↓ | | Energy invariance (eV) | |
|---|---|---|---|---|---|
| | | ID | Test | 3D Rotation ↓ | Reflection ↓ |
| SFA | 0 | **9.20** | **9.06** | $1.65 \cdot 10^{-3}$ | $1.76 \cdot 10^{-3}$ |
| Untrained SFA+MLP-SignNet | 0 | 11.3 | 11.2 | $2.12 \cdot 10^{-3}$ | $2.20 \cdot 10^{-3}$ |
| Trained SFA+MLP-SignNet | 454 | 10.5 | 10.7 | $1.60 \cdot 10^{-3}$ | $1.66 \cdot 10^{-3}$ |
| Untrained SFA+VN-SignNet | 0 | 9.41 | 9.40 | $1.28 \cdot 10^{-3}$ | $1.37 \cdot 10^{-3}$ |
| Trained SFA+VN-SignNet | 2,620 | 10.1 | 10.2 | $1.33 \cdot 10^{-3}$ | $1.40 \cdot 10^{-3}$ |
| Untrained VN-Pointnet (2 hid.) | 0 | 10.4 | 10.3 | $\mathbf{1.19} \cdot 10^{-3}$ | $1.30 \cdot 10^{-3}$ |
| Trained VN-Pointnet (2 hid.) | 1,310 | 10.1 | 9.85 | $1.30 \cdot 10^{-3}$ | $1.44 \cdot 10^{-3}$ |
| Untrained VN-Pointnet (0 hid.) | 0 | 9.51 | 9.49 | $1.21 \cdot 10^{-3}$ | $\mathbf{1.24} \cdot 10^{-3}$ |
| Trained VN-Pointnet (0 hid.) | 24 | 11.4 | 11.5 | $1.64 \cdot 10^{-3}$ | $1.74 \cdot 10^{-3}$ |
| Untrained VN-DGCNN | 0 | 9.92 | 9.94 | $1.32 \cdot 10^{-3}$ | $1.46 \cdot 10^{-3}$ |
| Trained VN-DGCNN | 663,804 | 9.34 | 9.25 | $1.79 \cdot 10^{-3}$ | $1.79 \cdot 10^{-3}$ |

*Table 9.* Equivariance and performance comparison of canonicalization methods on QM9 dataset for the target property $U_0$ (internal energy at 0 Kelvin). The FAENet backbone for this task has 6,495,127 parameters (5 interaction blocks). We measure the energy rotation invariance as the difference in prediction between every samples D1 (of the ID val split) and D2 defined as a SO(3) transformation of D1, in eV. All models were trained for 300 epochs using Duval et al. (2023) config.

## A.5. Relaxations from S2EF model for IS2RE

The models used to run the experiments with the relaxation methods were trained on the 2M train split of the S2EF dataset from OC20. This dataset has been shown to converge to similar performances as the complete dataset, which is way larger and takes too long to train on (Gasteiger et al., 2022). We report in Table 10 the performances of these models, which might help interpret some of the results for the relaxation.

| Model | EwT ↓ | Force MAE ↓ | Forces cos ↑ |
|---|---|---|---|
| FAENet with SFA | 10.7 | 0.044 | 0.32 |
| FAENet with Untrained PointNet | 10.5 | 0.0043 | 0.33 |
| FAENet without SFA | 10.0 | 0.042 | 0.34 |
| SchNet Base | 5.1 | 0.061 | 0.07 |

*Table 10.* Performance comparison of models on OC20 S2EF dataset on the VAL-ID split. The energy within threshold, Force MAE, and cos similarity are reported for these S2EF models that are then used for relaxations. Note that this method yields way longer training and inference times when compared to direct IS2RE as reported.

# B. Graph creation study

## B.1. Cutoff

The cutoff defines the distance within which a link between two atoms is created. All atoms that are at a distance smaller than this cutoff will be linked. However, in order to avoid cluttering the created graphs, most methods impose a maximum number of neighbors for every atom. This parameter is usually taken around 40 neighbors.

| Model | ID | | OOD-ADS | | OOD-CAT | | OOD-BOTH | |
|---|---|---|---|---|---|---|---|---|
| | EwT (%) ↑ | MAE (eV) ↓ | EwT (%) ↑ | MAE (eV) ↓ | EwT (%) ↑ | MAE (eV) ↓ | EwT (%) ↑ | MAE (eV) ↓ |
| Cutoff 30 - Max neighbours 40 | 2.65 | 0.697 | 1.45 | 0.906 | 2.86 | 0.691 | 1.53 | 0.846 |
| Cutoff 20 - Max. neighbours 40 | 3.08 | 0.673 | 1.85 | 0.808 | 2.86 | 0.669 | 1.86 | 0.757 |
| Cutoff 20 - Max. neighbours 10 | 2.25 | 0.768 | 1.51 | 0.988 | 2.52 | 0.754 | 1.38 | 0.928 |
| Cutoff 10 - Max. neighbours 50 | 4.17 | 0.553 | 2.81 | 0.640 | 4.12 | 0.551 | 3.02 | 0.585 |
| Cutoff 10 - Max. neighbours 40 | 4.29 | 0.555 | 2.95 | 0.631 | 4.33 | 0.553 | 2.71 | 0.587 |
| Cutoff 10 - Max. neighbours 30 | 4.43 | 0.551 | 2.65 | 0.655 | 4.51 | 0.552 | 2.51 | 0.611 |
| Cutoff 10 - Max. neighbours 20 | 4.38 | 0.551 | 2.46 | 0.676 | 4.45 | 0.551 | 2.55 | 0.621 |
| Cutoff 10 - Max. neighbours 10 | 4.49 | 0.553 | 2.84 | 0.627 | 4.34 | 0.549 | 3.01 | 0.582 |
| Cutoff 6 - Max. neighbours 40 | 4.31 | 0.553 | 3.00 | 0.626 | 4.39 | 0.554 | 2.81 | 0.577 |
| Cutoff 1 - Max. neighbours 40 | 1.35 | 1.069 | 1.32 | 1.112 | 1.33 | 1.051 | 1.37 | 1.018 |

*Table 11.* Impact of the cutoff on the performances of FAENet on the OC20 IS2RE task. Full table on all validation splits.

## B.2. Ewald-based Long-Range Message Passing

The main idea behind Ewald summation used in section 3.2 is to decompose the electrostatic interaction potential with a given charge into a short-range interaction and a long-range interaction term. The short-range contribution can be computed with real spatial features and the long-range contribution is computed using a Fourier transform. This principle is illustrated in Figure 3. This allows for computational methods in electrostatics to converge faster and with higher accuracy because the long-range interaction becomes more tractable.

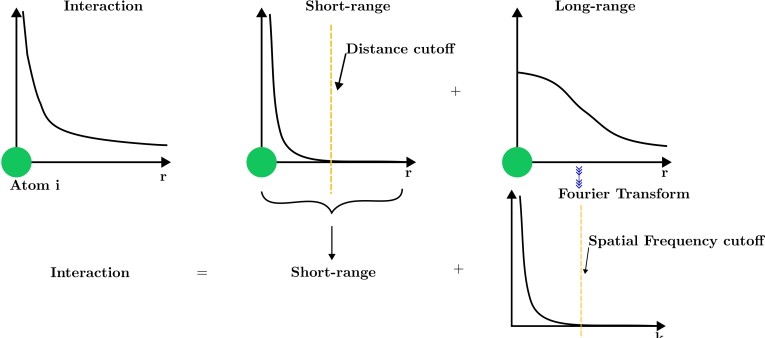

*Figure 3.* Ewald Summation interactions. The interaction term (left) is the result of the short-range interaction (middle) and the long-term interaction (right) which are both computed using cutoffs on respectively the real and the Fourier space. Adapted from (Kosmala et al., 2023)

In the case of GNNs, the short-range interaction is already computed by the currently implemented interaction blocks using a distance cutoff between the atoms, which omits the negligible parts of this interaction for further atoms. However, the heavy-tail of long-range interactions is then reported to a new term in real space, which doesn't diverge anymore for closer atoms. It can then be computed using the same cutoff idea but in the Fourier space where a *spatial frequency cutoff* is used to make it tractable as introduced in Kosmala et al. (2023).

**Periodic case.** In the case where there exists a spatial periodic tiling of materials (OC20 for example), it is possible to define the set of periodic cells localization $\Lambda = \{\lambda_1 \boldsymbol{v_1}, \lambda_2 \boldsymbol{v_2}, \lambda_3 \boldsymbol{v_3} \mid (\lambda_1, \lambda_2, \lambda_3) \in \mathbb{Z}^3\}$, where $\boldsymbol{v_1}, \boldsymbol{v_2}, \boldsymbol{v_3}$ define the periodic cell lattice, similarly to the periodic interval in the 1D case. In the real space, the long-range interaction component would be written as a sum over all of the elements on the infinite tiling, which can be decomposed as a Fourier series

expansion using the reciprocal lattice $\Lambda'$. This reciprocal lattice would be similar to the $2\pi$ in the 1D case of the Fourier transform. It is the periodic space of all the wavevectors of the Fourier series. This results in the proposed expansion for Ewald message passing:

$$M^{lr}(x_i) = \sum_{\boldsymbol{k} \in \Lambda'} \exp\left(i\boldsymbol{k}^T x_i\right) \cdot \sum_{j \in \mathcal{S}} h_j \exp\left(-i\boldsymbol{k}^T x_j\right) \cdot \hat{\Phi}^{lr}(\|\boldsymbol{k}\|), \tag{5}$$

where $M^{lr}(x_i)$ corresponds to the long-range message computed at node $i$ from all of the nodes in the system $S$, and $\hat{\Phi}^{lr}$ is a learned Fourier coefficient of a radial basis function representing the interaction. The cutoff in the Fourier basis $c_k$ is then set to make the sum finite over the set $\{k \in \Lambda', \|k\| \leq c_k\}$. Since the number of wavevectors used for the computation is finite, $\hat{\Phi}(\|\boldsymbol{k}\|)$ is learned for every $\boldsymbol{k}$. Since Ewald summation applies to periodic structures, the authors of Ewald message passing Kosmala et al. (2023) propose tricks to deal with the aperiodic case by assuming an infinite tiling.

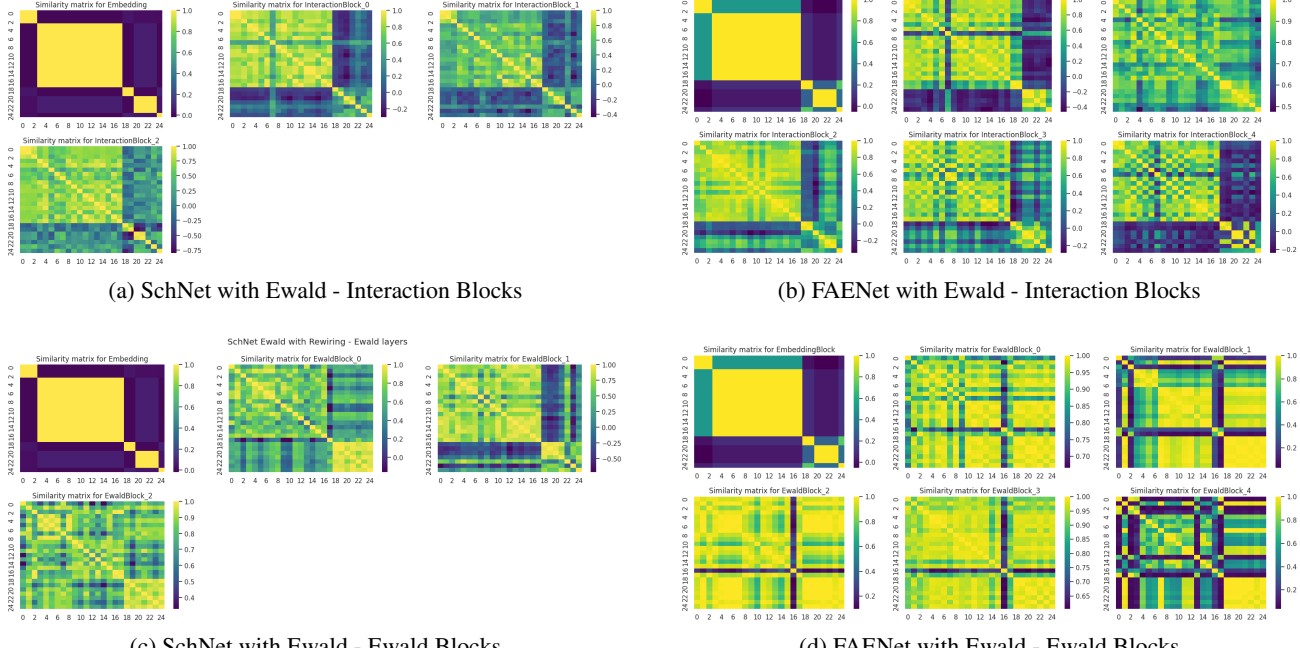

(a) SchNet with Ewald - Interaction Blocks

(b) FAENet with Ewald - Interaction Blocks

(c) SchNet with Ewald - Ewald Blocks

(d) FAENet with Ewald - Ewald Blocks

*Figure 4.* Similarity matrix of the embeddings of the atoms of a system for different interaction blocks on FAENet and SchNet with Ewald message passing on the models. The visualized layers here are the standard interaction blocks in the first row and the Ewald interaction blocks in the second row. The two outputs are summed to get the final representation for Ewald shown in Figure 1.

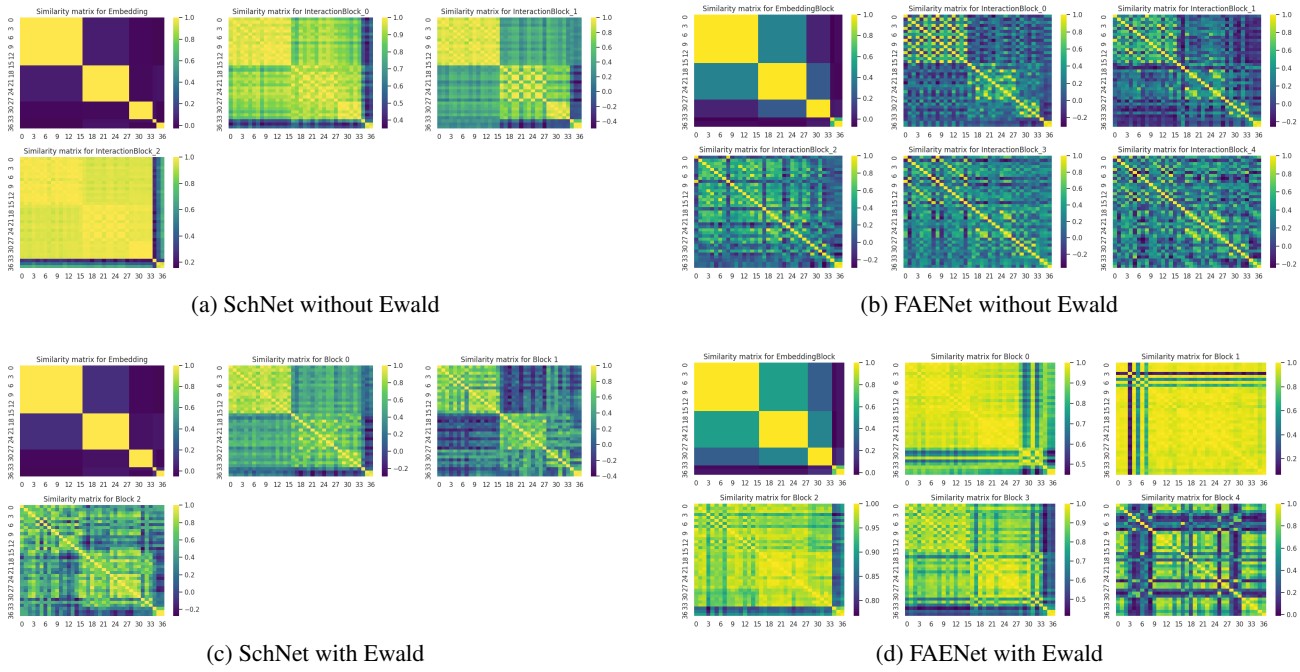

(a) SchNet without Ewald

(b) FAENet without Ewald

(c) SchNet with Ewald

(d) FAENet with Ewald

*Figure 5.* Same plots as Figure 1 but with a second randomly picked system from the OC20 train split.

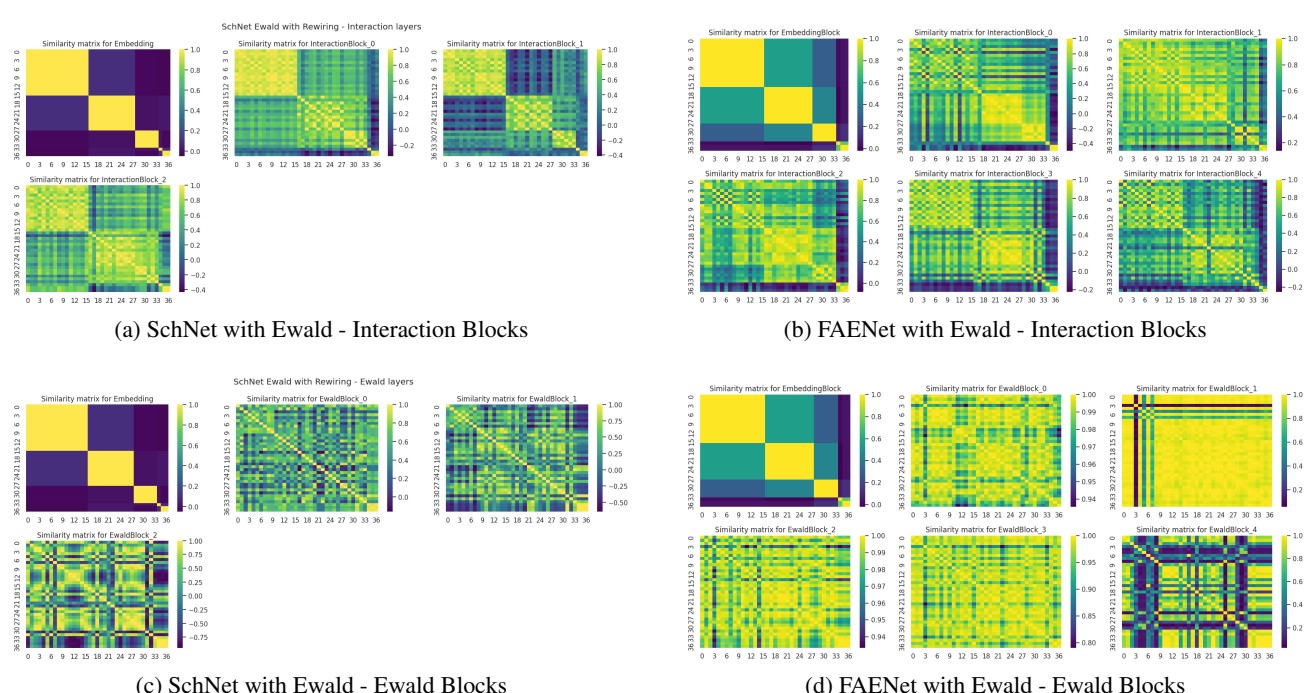

(a) SchNet with Ewald - Interaction Blocks

(b) FAENet with Ewald - Interaction Blocks

(c) SchNet with Ewald - Ewald Blocks

(d) FAENet with Ewald - Ewald Blocks

*Figure 6.* Same plots as Figure 4 but with a second randomly picked system from the OC20 train split.

| Model | ID | | OOD-ADS | | OOD-CAT | | OOD-BOTH | |
|---|---|---|---|---|---|---|---|---|
| | EwT (%) ↑ | MAE (eV) ↓ | EwT (%) ↑ | MAE (eV) ↓ | EwT (%) ↑ | MAE (eV) ↓ | EwT (%) ↑ | MAE (eV) ↓ |
| FAENet (Graph Rewiring) | 4.05 | 0.551 | 2.65 | 0.650 | 4.29 | 0.550 | 2.76 | 0.601 |
| FAENet (Graph Rewiring) + Ewald | 4.12 | 0.562 | 2.68 | 0.648 | 4.14 | 0.563 | 2.83 | 0.597 |
| FAENet (No Graph Rewiring) | 4.54 | 0.544 | 2.59 | 0.657 | 4.66 | 0.539 | 2.65 | 0.601 |
| FAENet (No Graph Rewiring) + Ewald | 4.11 | 0.556 | 2.75 | 0.626 | 4.13 | 0.553 | 2.85 | 0.569 |
| SchNet (Graph Rewiring) | 3.18 | 0.641 | 2.53 | 0.720 | 3.00 | 0.638 | 2.59 | 0.642 |
| SchNet (Graph Rewiring) + Ewald | 3.54 | 0.604 | 2.53 | 0.665 | 3.53 | 0.599 | 2.67 | 0.608 |
| SchNet (No Graph Rewiring) | 2.93 | 0.654 | 2.22 | 0.700 | 3.04 | 0.646 | 2.54 | 0.656 |
| SchNet (No Graph Rewiring) + Ewald | 3.48 | 0.597 | 2.76 | 0.647 | 3.56 | 0.599 | 2.73 | 0.612 |

*Table 12.* Energy prediction errors with and without Ewald Message Passing. Graph Rewiring refers to removing the subsurface atoms from the system (Duval et al., 2022). In this table, FAENet is taken with 5 interaction layers (top config), while SchNet uses 3 interaction layers.

| Model | MAE (meV) ↓ | MSE ((meV)$^2$) ↓ |
|---|---|---|
| FAENet | 8.44 | 0.600 |
| FAENet + Ewald | 8.34 | 0.574 |
| SchNet | 16.0 | 1.18 |
| SchNet + Ewald | 11.5 | 0.73 |

*Table 13.* Energy prediction errors with and without Ewald Message Passing on the test split of the QM9 dataset for the target property $U_0$ (Internal energy at 0K).

## C. Noisy Nodes

### C.1. Experimental setup

Runs are done on a single Nvidia Quadro RTX 8000 GPU with 48 GB memory or, if indicated, on a single Nvidia A100 GPU with 80 GB memory.

### C.2. Noisy Nodes implementation

In practice, we perturb the input node positions of the graph $G$ with a noise $\sigma$ and train the model with two loss terms, a Noisy Nodes loss term and the primary loss (associated with the main task) term

$$\mathcal{L} = \lambda \cdot \mathcal{L}_{NNodes}(\hat{G}', V') + \mathcal{L}_{Primary}(\hat{G}', V'), \qquad (6)$$

where $\lambda$ is the weight we assign to the auxiliary denoising task, $\hat{G}' = \tilde{\Phi}(\tilde{G})$ is the output of the model $\tilde{\Phi}$, $\tilde{G}$ is the noised graph, and $V'$ can either be the target nodes features (e.g. atom positions at equilibrium, that is the IS2RS task) or the initial nodes positions (see next subsection on denoising pre-training).

For the input, we first interpolate between initial structure and relaxed structure and then add Gaussian noise, that is for each node $i$, the input positions of the input "Noisy Nodes graph" $\tilde{x}^i$ are defined as

$$\tilde{x}^i = \begin{cases} \gamma(x_{rel}^i - x_{init}^i) + Z^i & \text{with probability } 0.5 \\ x_{init}^i & \text{with probability } 0.5, \end{cases} \qquad (7)$$

with random interpolation factor $\gamma \sim U[0, 1]$ independent between graphs and iid Gaussian noises $Z^i \sim \mathcal{N}(0, \sigma)$ with $\sigma = 0.3$. The Noisy Nodes target is $\Delta_{pos}^i = x_{rel}^i - \tilde{x}^i$ and therefore the auxiliary loss term is $\|\Delta_{pos}^i - \Phi(\tilde{x}^i)\|_1$ for the model $\Phi$. Our total loss is the sum of the energy MAE loss and the auxiliary loss weighted by a number $\lambda$, as in Equation 6. Both the primary loss and Noisy Nodes Loss (before multiplication by $\lambda$) typically have the same value between 1 and 2.5.

For the IS2RE training, we add a position decoding head to the preexisting energy prediction head. The position decoding head, as with the force prediction head introduced in (Duval et al., 2023), is a 2-layer MLP with Batchnorm.

### C.3. Related work

Since Noisy Nodes performs denoising as an auxiliary task during training, the representation learning benefits of denoising are limited to the downstream dataset on which the model is trained. Zaidi et al. (2022) propose to rather perform denoising

as a pre-training objective on a large, unlabelled dataset of atomic structures.

Shoghi et al. (2024) introduce Joint Multi-domain Pre-training (JMP), a supervised pre-training strategy that simultaneously trains on various datasets from different chemical domains (OC20 (Chanussot et al., 2020), OC22 (Tran et al., 2023), ANI-1x (Smith et al., 2020), and Transition-1x (Schreiner et al., 2022)), treating each dataset as a unique pre-training task within a multi-task framework.

### C.4. Training hyperparameters

The most obvious changes to the training hyperparameters that should theoretically allow to leverage the denoising auxiliary task are to increase the depth of the network and the number of epochs. First, we observe that we reach convergence on the validation set for the energy MAE more slowly when using the auxiliary task because of a more complex loss to minimize and a higher number of model weights (due to the supplementary interaction blocks). Hence, unless otherwise stated, the number of epochs is 50 in the IS2RE with IS2RS auxiliary task experiments.

(Liao & Smidt, 2023) use a linearly decaying weight from 15 to 1 for the auxiliary IS2RS loss to encourage the model to learn more from the auxiliary task in the beginning but focus on the primary task at the end of training. We also tested a constant auxiliary weight of 1, a weight decaying from 30, and cosine annealing with a linear warmup scheduler, but this yielded worse or equivalent results. Thus, unless otherwise stated, we always use as an auxiliary weight scheduler the one of Equiformer in the following IS2RE with IS2RS auxiliary task experiments.

Since we always use the MAE as energy loss for the IS2RE main task, it proves essential to use the MAE loss for the auxiliary position loss to leverage the benefits of Noisy Nodes. Indeed, our experiments using the MSE position loss led to a collapse of the node embeddings at the last interaction layers, that could be observed by plotting the MAD throughout the interaction layers.

### C.5. Results

#### C.5.1. COMPARING DEPTH FOR CLASSICAL FAENET

In the results of Table 14, the number of warmup steps is 6000 for about 180k steps. The batch size is 128, hence the lower throughput than the one with the same configs of (Duval et al., 2023). In these runs, the hyperparameters are the top configs of (Duval et al., 2023) except for slight differences: the number of hidden channels in the embedding blocks is a bit lower. Also, contrary to the top configs of (Duval et al., 2023), we train for 50 epochs (instead of 12) with no early stopping to be in a comparable setup to the IS2RE with IS2RS auxiliary task models.

| Interaction blocks | Parameters (millions) | Time Train ↓ | Infer. ↑ | Energy MAE (meV) ↓ ID | OOD Ads | OOD Cat | OOD Both | Average | EwT (%) ↑ ID | OOD Ads | OOD Cat | OOD Both | Average |
|---|---|---|---|---|---|---|---|---|---|---|---|---|---|
| 5 | 5.9 | 19min | 786 | 556 | 685 | 552 | 636 | 607 | 4.41 | 2.26 | 4.51 | 2.47 | 3.41 |
| 8 | 9.2 | 25min | 676 | 554 | 643 | 558 | 596 | **588** | 4.31 | 2.66 | 4.40 | 2.75 | 3.53 |
| 10 | 11.4 | 29min | 621 | 552 | 649 | 554 | 603 | 590 | 4.56 | 2.77 | 4.20 | 2.72 | **3.56** |
| 12 | 13.7 | 34min | 527 | 551 | 661 | 556 | 609 | 594 | 4.01 | 2.57 | 4.21 | 2.53 | 3.33 |
| 14 | 15.9 | 37min | 498 | 621 | 749 | 608 | 693 | 668 | 3.42 | 2.06 | 3.27 | 2.31 | 2.77 |
| 16 | 18.2 | 40min | 457 | 590 | 704 | 592 | 647 | 633 | 4.05 | 2.51 | 3.90 | 2.53 | 3.25 |

*Table 14.* Comparison varying the number of interaction blocks for FAENet without auxiliary task with the top configs of (Duval et al., 2023) except for slight differences. Scalability is measured with training time for one epoch (Train, in minutes) and inference throughput (Infer., number of samples processed in a second). The best score is in bold and the second-best score is underlined. Performances are slightly worse than the best performances presented in (Duval et al., 2023) because of different hyperparameters, but the main conclusion is that the performances worsen when we add interaction layers after 12.

#### C.5.2. COMPARING DEPTH FOR FAENET WITH NOISY NODES IS2RS AUXILIARY TASK

In Table 15, we observe a clear positive correlation between the depth of the model and its performances in terms of energy MAE and EwT. The gain in performance is very clear between 5 and 16 interaction layers, then increases much more slowly up to 28 layers.

| Interaction blocks | Parameters (millions) | Time | | Energy MAE (meV) ↓ | | | | | EwT (%) ↑ | | | | |
|---|---|---|---|---|---|---|---|---|---|---|---|---|---|
| | | Train ↓ | Infer.↑ | ID | OOD Ads | OOD Cat | OOD Both | Average | ID | OOD Ads | OOD Cat | OOD Both | Average |
| 5 | 4.2 | 17min | 724 | 523 | 592 | 525 | 545 | 546 | 4.59 | 2.92 | 4.40 | 3.15 | 3.76 |
| 8 | 6.6 | 22min | 679 | 518 | 600 | 526 | 561 | 551 | 5.13 | 2.88 | 5.21 | 2.74 | 3.99 |
| 10 | 8.1 | 27min | 584 | 513 | 606 | 521 | 561 | 550 | 5.17 | 2.82 | 5.03 | 2.81 | 3.96 |
| 12 (A100) | 9.7 | 30min | 504 | 517 | 589 | 523 | 546 | 544 | 5.10 | 3.17 | 4.92 | 3.04 | 4.06 |
| 14 | 11.2 | 33min | 525 | 507 | 580 | 519 | 541 | 537 | 5.20 | 3.13 | 5.21 | 3.25 | 4.20 |
| 16 | 12.8 | 35min | 465 | 505 | 566 | 516 | 527 | 529 | 5.17 | 3.63 | 5.00 | 3.69 | 4.37 |
| 18 | 14.3 | 38min | 446 | 508 | 596 | 518 | 554 | 544 | 5.31 | 3.22 | 5.58 | 2.97 | 4.27 |
| 20 | 15.9 | 42min | 423 | 508 | 593 | 513 | 553 | 542 | 5.39 | 3.16 | 5.18 | 2.88 | 4.15 |
| 22 | 17.4 | 45min | 401 | 507 | 569 | 515 | 528 | 530 | 5.24 | 3.26 | 5.06 | 3.32 | 4.22 |
| 24 | 19.0 | 48min | 375 | 500 | 574 | 510 | 534 | 529 | 5.70 | 3.43 | 5.43 | 3.31 | **4.47** |
| 26 | 20.6 | 50min | 335 | 504 | 562 | 512 | 521 | **525** | 5.15 | 3.68 | 5.20 | 3.67 | 4.43 |
| 28 | 22.1 | 56min | 349 | 505 | 567 | 513 | 517 | **525** | 5.20 | 3.64 | 5.27 | 3.28 | 4.35 |

*Table 15.* Comparison of number of interaction blocks for IS2RE with auxiliary IS2RS. Scalability is measured witht raining time for one epoch (Train, in minutes) and inference throughput (Infer., number of samples processed in a second). Here the canonicalization technique is SE(3)-SFA, the number of warmup steps is 192000 (about half total number of steps), and the batch size is 64. Best score is in bold, second best score is underlined.

### C.5.3. CANONICALIZATION COMPARISON

We use SE(3)-SFA because it is less equivariant to reflections than Stochastic FA (which samples a frame in E(3)), but it has to learn data symmetries from fewer frames, which helps training. For both SE(3)-SFA and No-FA models, we observe in Table 16 that the equivariance-invariance do not seem to be correlated to the number of interaction layers. Moreover, we do not observe a clear correlation between the Average energy MAE and the equivariance-invariance.

When using SE(3)-SFA, we see a clear correlation between the increase in the number of interaction layers and the performances. In the No-FA case, the impossibility to learn equivariance and invariance even when adding more layers to the model might account for the less clear positive correlation between the performances (in terms of energy MAE) and the number of interaction layers than when we were applying SE(3)-SFA.

| Canonicalization | Interactions | 2D E-RI ↓ | 2D E-Refl-I ↓ | 2D Pos-RI ↓ | 2D Pos-Refl-I ↓ | Average E-MAE (meV) ↓ |
|---|---|---|---|---|---|---|
| SE(3)-SFA | 5 | 22.7 | 34.7 | 55.0 | 80.3 | 546 |
| SE(3)-SFA | 14 | 29.7 | 43.2 | 61.7 | 90.4 | 537 |
| SE(3)-SFA | 16 | 21.4 | 33.6 | 56.0 | 83.4 | 529 |
| SE(3)-SFA | 18 | 19.0 | 30.2 | 53.9 | 78.4 | 544 |
| SE(3)-SFA | 20 | 22.6 | 34.7 | 56.2 | 82.8 | 542 |
| SE(3)-SFA | 22 | 20.8 | 32.6 | 56.3 | 81.7 | 530 |
| SE(3)-SFA | 28 | 25.5 | 37.3 | 60.8 | 86.5 | **525** |
| SE(3)-SFA no aux | 5 | 6.9 | 8.9 | | | 569 |
| No-FA | 8 | 111 | 107 | 253 | 238 | 561 |
| No-FA | 10 | 110 | 107 | 251 | 233 | 577 |
| No-FA | 14 | 121 | 117 | 275 | 254 | 562 |
| No-FA | 18 | 115 | 111 | 273 | 257 | 608 |
| No-FA | 22 | 116 | 111 | 260 | 240 | **554** |
| No-FA | 26 | 120 | 117 | 288 | 271 | 579 |

*Table 16.* Comparison of using SE(3)-SFA on both the energy and position prediction heads to No FA and to the top configs of FAENet with no auxiliary task from (Duval et al., 2023). The symmetry-preservation metrics are in meV for the energy and milli-Angstroms for the positions. The best model for each of the 2 categories is in bold and the second best is underlined. We do not seem to have a correlation between the quality of the invariance and equivariance with the number of interaction layers in both categories.

### C.6. Pre-tranining on different tasks

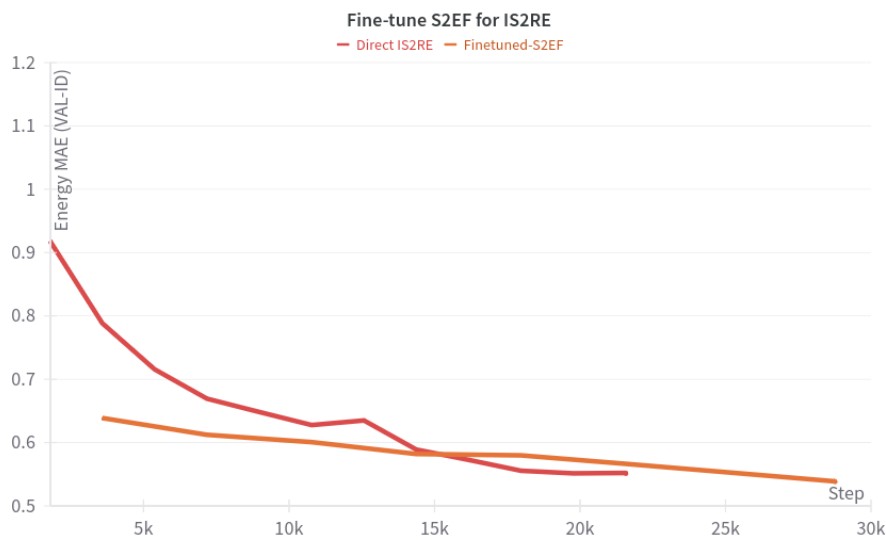

*Figure 7.* Validation curve during training for both a model trained directly from scratch for IS2RE and an S2EF model fine-tuned on IS2RE