# OpenReview forum: "Improving Molecular Modeling with Geometric GNNs: an Empirical Study"
_ICML.cc/2024/Workshop/ML4LMS — ML4LMS Poster_

### Official Review · Reviewer_vdZK · 2024-05-31
**An insightful and comprehensive study of geometric graph neural networks for atomic (materials) systems**

**Rating:** 8
**Confidence:** 4

**Review:**

**Summary:** The authors present a comprehensive empirical analysis of the design space of geometric graph neural networks (GNNs) for atomic (materials) systems.

**Strengths and Weaknesses:**

Points of strength:
- The authors' study provides many actionable and insightful findings for future work in this domain:
  1. Approximate equivariance constraints (e.g., through stochastic frame averaging) seem to outperform strict equivariance constraints in the context of geometric atomic GNNs
  2. Physics-informed message passing mechanisms seem to primarily benefit expressivity-limited GNNs such as SchNet more than more expressive models such as FAENet.
  3. Auxiliary denoising tasks such as Noisy Nodes improve performance consistently but must be incorporated into existing atomic GNNs carefully to avoid negative interplay with the primary learning objective(s).
- Taken together, these above insights should prove quite useful for the field.

Points for improvement:
- I would recommend the authors perform repeat experiments (repeating the standard deviation of each metric) for the paper's primary experiments. I understand that this is not reasonable to do for every experiment included in the paper, but for the experiments most important to the main claims of this paper, error bars should be included to give readers confidence that this paper's findings will hold up in follow-up studies.

**Recommendation:** Given the insightful direction and results of this research and its relevance to this workshop's interests, I very much recommend this work for acceptance.

**Rationale behind Recommendation:** The authors propose a quite comprehensive analysis of GNNs for atomic (materials) systems that outlines important future research directions in this domain.

**Questions:**
(1) Regarding the authors' experiments investigating the (AlphaFold 3-inspired) question of whether equivariance is truly necessary, have the authors also considered examining this question from the perspective of the learning objective differences between predictive machine learning tasks and *generative* machine learning tasks? In this work, the authors primarily study predictive tasks, whereas for methods such as AlphaFold 3, generative modeling (and its associated loss functions, especially including domain-specific bond penalties) may be compensating (e.g, essentially filling in the gap) for equivariance constraints, in addition to vast data augmentations that AlphaFold 3 employs during training (i.e., 48 random roto-translations of an input system per forward pass).

**Feedback:**
- Vague wording: "the graph creation part" on Line 016
- Vague wording: "systems structures" on Line 034
- Missing abbreviation: Please define "CN" on Line 085 before it is first referenced in the paper
- Typo: "Laslty" on Line 087
- Vague wording: "quasi no differences" on Line 088
- Vague wording: "on FAENet" on Line 119
- Vague wording: "propagated messages propagated" on Line 142
- Vague wording: "80GB A100 GPU the rewiring" on Line 143
- Typo: "that go back to" on Line 233
- Typo: "via auxiliary task or other datasets" on Line 234

**Submission Type:** The authors' submission successfully complies with the corresponding formatting requirements to the best of my knowledge.

---

### Official Review · Reviewer_d1Er · 2024-06-02
**Borderline paper, but with possibility of triggering relevant discussions**

**Rating:** 6
**Confidence:** 4

**Review:**

In this work, authors intend to perform an empirical study on canonicalization methods, graph creation strategies, and auxiliary tasks, such that their results can guide other researchers when choosing the right approach for their needs. They use the recent unconstrained FAENet as the backbone model, and focus on the IS2RE task of the Open Catalyst project.

I have the impression that a discussion and a comparison of canonicalization methods is necessary, given the current debate around unconstrained models. However, authors are using a single model and a single primary task for their study, falling behind an ‘empirical study on Geometric Graph Neural Networks for 3D atomic systems’ and ‘helping [researchers] quickly choose the right modeling component’. I would frame the authors’ work in a narrower domain than the full richness of geometric graph neural networks. On the other hand, ablations are quite interesting (for example the incorporation of Ewald), but they seem disconnected from a single main narrative (canonicalization, maximum number of neighbors, graph rewiring, noisy nodes…)

I propose to (borderline) accept the paper because of the discussions it can trigger, which are important, and because both the authors and the community might benefit from them.

---

### Official Review · Reviewer_eVcF · 2024-06-11
**Review of Submission120**

**Rating:** 7
**Confidence:** 4

**Review:**

This paper investigates the impact of various modeling choices in Geometric GNNs for 3D atomic systems. The study covers three key areas: canonicalization methods, graph creation strategies, and auxiliary tasks, aiming to provide insights to help researchers select optimal modeling components for molecular modeling tasks. Overall, this paper covers a broad range of factors and is well-structured. Here are a few comments/questions to address:

1. The paper primarily focuses on the OC20 dataset. The authors may consider conducting the same experiments on additional datasets to strengthen the arguments.

2. While the empirical results are valuable, the paper could benefit from some theoretical explanations on why certain methods outperform the others.